# Influence of Manufacturing Process on the Microstructure, Stability, and Sensorial Properties of a Topical Ointment Formulation

**DOI:** 10.3390/pharmaceutics15092219

**Published:** 2023-08-28

**Authors:** Pui Shan Chow, Ron Tau Yee Lim, Febin Cyriac, Jaymin C. Shah, Abu Zayed Md Badruddoza, Thean Yeoh, Chetan Kantilal Yagnik, Xin Yi Tee, Annie Bao Hua Wong, Vernissa Dilys Chia, Guan Wang

**Affiliations:** 1Institute of Sustainability for Chemicals, Energy and Environment, Agency for Science, Technology and Research (A*STAR), 1 Pesek Road, Jurong Island, Singapore 627833, Singapore; ann_chow@isce2.a-star.edu.sg (P.S.C.); feb.cyriac@gmail.com (F.C.); chetan_kantilal@isce2.a-star.edu.sg (C.K.Y.); tee_xin_yi@isce2.a-star.edu.sg (X.Y.T.); annie_wong@isce2.a-star.edu.sg (A.B.H.W.); wangg@isce2.a-star.edu.sg (G.W.); 2Drug Product Design, Worldwide Research, Development and Medical, Pfizer Inc., Groton, CT 06340, USA; abuzayedmd.badruddoza@pfizer.com (A.Z.M.B.); yeoh@yahoo.com (T.Y.)

**Keywords:** ointment, microstructure, rheology, sensorial, manufacturing process, design of experiments, topical formulation

## Abstract

The manufacturing process for ointments typically involves a series of heating, cooling, and mixing steps. Precise control of the level of mixing through homogenization and the cooling rate, as well as temperature at different stages, is important in delivering ointments with the desired quality attributes, stability, and performance. In this work, we investigated the influence of typical plant processing conditions on the microstructure, stability, and sensorial properties of a model ointment system through a Design of Experiments (DoE) approach. Homogenization speed at the cooling stage after the addition of the solvent (propylene glycol, PG) was found to be the critical processing parameter that affects stability and the rheological and sensorial properties of the ointment. A lower PG addition temperature was also found to be beneficial. The stabilization of the ointment at a lower PG addition temperature was hypothesized to be due to more effective encapsulation by crystallizing mono- and diglycerides at the lower temperature. The in vitro release profiles were found to be not influenced by the processing parameters, suggesting that for the ointment platform studied, processing affects the microstructure, but the effects do not translate into the release profile, a key performance indicator. Our systematic study represents a Quality-by-Design (QbD) approach to the design of a robust manufacturing process for delivering stable ointments with the desired performance attributes and properties.

## 1. Introduction

The typical ointment formulation is an oil-based formulation with a semi-solid texture comprising a solvent phase dispersed in an oil phase stabilized by emulsifying agents. A drug or an active ingredient can be dissolved in the solvent phase and dispersed in the external oil phase. The typical oil phase of ointment comprises petrolatum and stiffening agents and/or emulsifying agents such as glyceride derivatives that provide the desired stiffness/strength to keep the solvent phase dispersed. In addition, to provide emollience for skin, mineral oil is typically used in ointments [1].

The manufacturing process for ointments involves heating to melt the petrolatum and waxes, followed by addition of other liquid components, such as polar solvents, with mixing/homogenization for uniform dispersion. The ointment is then cooled to room temperature before filling into containers such as jars and tubes. It is anticipated that homogenization should affect the distribution or dispersion of the various components at the different stages, and both heating and cooling, along with homogenization, affect the distribution and crystallization of emulsifying wax and stiffening agents around the solvent droplets and the oil phase. Although the term microstructure has been recently used to characterize semi-solid topical products such as ointments, its definition is not clearly articulated and agreed upon [2]. The microstructure of a dispersed multi-phasic system, such as ointments, is essentially a representation of how various components or excipients are arranged and interact with each other in a structure that influences important properties of the system. Hence, microstructure is typically referred to as the droplet size of dispersed liquids, crystalline material distribution around the droplet and in the bulk, and rheological or viscoelastic properties of the system. These microstructural properties have an impact on the physical stability of the ointment as a result of growth in droplet size and, eventually, phase separation. Heating to melt the petrolatum and waxes, followed by mixing/homogenization, affects the distribution of the various components at the different stages. Both heating and cooling, along with homogenization, affect the distribution and crystallization of emulsifying wax and stiffening agents around the solvent droplets and the oil phase. Therefore, processing conditions should impact the microstructure of the formulation, which, in turn, affects the drug product (DP) stability as well as the performance attributes, such as release and skin permeation.

Besides influencing the active release and skin permeation behavior, the microstructural properties—including droplet size, structure, and distribution of the stiffening and emulsifying agents around the oil droplet—also affect the sensorial properties as the product is applied to skin. In addition, congealing of petrolatum and distribution of the waxes and stiffening agents as they crystallize would generate a viscoelastic system with different rheological properties. The rheological properties would influence the phase separation of dispersed solvent due to various de-mixing phenomena such as coalescence, sedimentation, and/or creaming, and potentially impact the bio-performance, i.e., in vitro release and skin permeation. While the in vitro release and permeation profile of a drug (flux) influences drug delivery and hence efficacy, sensorial properties are also an important concern, as they affect consumer perception and acceptance of the product.

The objective of this study was, therefore, to study the influence of typical plant processing conditions of various unit processes on the microstructure, stability, and sensorial properties of a topical ointment formulation through a systematic DoE approach. The focus of the current study is the effect of processing on the microstructure of the ointment formulation. However, if significant impacts of process conditions on microstructure are observed, the second study would involve an assessment of the performance attributes, such as API release.

## 2. Materials and Methods

### 2.1. Materials

Both the white petrolatum and paraffin wax used in this study were Ph. Eur grades manufactured by H&R Gruppe (Hamburg, Germany) and VWR International (Singapore), respectively. Mono-and-Diglycerides (Geleol™) was purchased from Gattefosse (Saint Priest, France). Propylene glycol (Ph. Eur grade) was purchased from Thermo Fisher Scientific (Loughborough, UK). The active pharmaceutical ingredient, lidocaine, was purchased from Sigma Aldrich (Singapore). All the ingredients were used directly as supplied for ointment preparation.

### 2.2. Ointment Preparation Procedure

Table 1 shows the ointment composition used in this study. It consists of white petrolatum (WP) and paraffin wax (PW) as the primary phase. Mono-and-Diglycerides (MDG) primarily served as the emulsifying agent for stabilizing propylene glycol (PG) droplets, the active phase dispersed in the primary or continuous phase. MDG are widely used as an emulsifying agent to stabilize topical formulations, including ointments [1,3,4,5,6,7,8].

High-speed mixing emulsifying equipment (ESCO EL1 ECO, ESCO-Labor AG, Riehen, Switzerland) consisting of a glass-jacketed vessel equipped with a high-speed shearing homo-mixer and a scraper/paddle mixer was used as the representative manufacturing container for ointments. The equipment was capable of manufacturing semi-solid products under vacuum. The temperature was controlled by a water circulator (Julabo FP50, JULABO GmbH, Seelbach, Germany) with an attached thermocouple (Julabo Pt100). The ointment was prepared in two stages, namely compounding and post compounding, as shown in the schematic in Figure 1. The ointment preparation steps are outlined below in more detail. The levels of each processing condition are listed in Table 2.

Compounding
Step 1:Charge white petrolatum, paraffin wax and Mono-and-Diglycerides into the jacketed mixing vessel and heat to 75 ± 5 °C while mixing until all materials are fully melted.Step 2:Transfer melted waxes into the main vessel and mix (90 rpm) at 75 ± 5 °C for at least 10 min.

In the following steps, [F_1_] to [F_4_] represent set points of various processing parameters/factors investigated for their effect on ointment properties with a statistically appropriate DoE study.
Step 3:The cooling rate is set at [F_1_] to cool the contents of the main vessel while mixing at 90 rpm. When the temperature reaches 50 ± 3 °C, homogenize at [F_2_] for 5 min while continuing mixing at 90 rpm until the temperature reaches [F_3_]. [F_1_], [F_2_], and [F_3_] are critical process parameters investigated for the effect on drug product quality attributes.Step 4:Dissolve lidocaine in propylene glycol. Charge propylene glycol into main vessel while mixing (90 rpm) at [F_3_]. After the addition is completed, homogenize at [F_2_] for 20 min at every 30 min interval. Homogenization is repeated twice for a total mixing and homogenization duration of 60 min.Step 5:The cooling rate is set at 0.5 ± 0.1 °C/min to cool the contents to 32 ± 3 °C while mixing (50 rpm). Homogenization at [F_4_] is applied for 5 min when the temperature reaches 36 ± 3 °C.

Post Compounding
Step 6:While mixing at 50 rpm, apply shear with homogenization at 1000 rpm for 10 min at 32 ± 3 °C.Step 7:Stop shearing. The product is cooled to 25 ± 2 °C at a cooling rate of 0.25 ± 0.1 °C/min while mixing at 10 rpm. Once the temperature reaches 25 ± 2 °C, mixing is stopped, and the product is held for 24 h before characterization is conducted.

### 2.3. Characterization Techniques

#### 2.3.1. Microscopy Analysis

Microscopic images of the ointments were acquired using an Olympus BX51 polarizing microscope (Olympus, Singapore) equipped with a Nikon DS-Fi3 high-resolution camera (Nikon, Singapore). The images were analyzed using the NIS-Elements AR imaging analysis software version 4.60.00 (Nikon, Singapore)). The size of the PG droplets and observed crystalline structures due to MDG were measured manually. At least 300 crystallites or droplets were measured to ensure statistical significance and that they were representative of the bulk product.

#### 2.3.2. Physical Stability

The physical stability of ointment samples processed with different processing parameters was characterized using a LUMiSizer (LUM GmbH, Berlin, Germany). It employs centrifugal force to accelerate creaming or settling due to density differences in the samples, presumably due to instability [9,10]. While the sample is subjected to centrifugation to accelerate phase separation, light transmitted through the sample is recorded as a function of time and position over the entire sample length. The instability analysis was performed using the SepView6.0 software (LUM GmbH, Berlin, Germany). An instability index (IS) is calculated by integrating the normalized transmitted light over the entire length of the sample [11]. The instability index is a dimensionless number in the range of 0 to 1, where 0 indicates a stable system and 1 an unstable system [12]. When a system is unstable and starts to phase separate under centrifugation, a clear separated/clarification phase starts to appear, and more light will be transmitted through the sample, resulting in a higher instability index. Therefore, a higher instability index indicates a larger extent or degree of phase separation and, hence, lower stability. The equipment parameters used for the measurement were set as follows: Sample (approximately 0.4–0.5 mL) was placed into a LUM Polycarbonate (PC) cell (2 mm), 4000 rpm; duration, 7.5 h; and temperature, 34 °C.

#### 2.3.3. Rheological Properties

Rheological characterization of ointment formulations was performed on an MCR 302 rheometer (Anton Paar GmbH, Graz, Austria). All measurements (sample size ~0.5–1.0 g) were performed at 25 ± 0.1 °C using a sand-blasted steel parallel plate (25 mm, Ra = 5.4 µm) at a measurement gap of 1 mm. A rest time of 10 min was employed after closing the gap to 1 mm by controlling the normal force to relieve the residual stress and to help build up the microstructure. Small Amplitude Oscillatory Shear (SAOS) measurements were performed at 1 Hz from a strain of 0.001 to 1000% without imposing a time constraint on the measurement duration. Linear viscoelastic properties such as storage modulus (G′), loss modulus (G″), complex viscosity (η*), etc. were obtained from SAOS. Additionally, the spreadability and cohesion strength of the formulations were also obtained using the parallel plate rheometer using a methodology described elsewhere [13]. All measurements were repeated thrice to ensure reproducibility.

#### 2.3.4. Sensorial Properties—Texture and Tribology Analysis

Texture analysis measurements were performed using a Brookfield CTX texture analyzer (AMETEK Brookfield, Middleboro, MA, USA). A steel hemispherical probe of 12.7 mm diameter was used to perform penetration tests on the samples [14]. The probe was allowed to penetrate the sample (50 mL) over a depth of 20 mm at a preset load. The probe then remained at the preset depth for about 1 s before it was retracted to its initial position. A speed of 0.25 mm/s was used during the downward and upward movement of the probe. The data were analyzed using TEXTURE PRO version 1.0 build 19 (AMETEK Brookfield, Middleboro, MA, USA) to obtain different textural attributes. The measurements were repeated thrice to check reproducibility.

As spreadability and cohesion measurements using the texture analyzer require a large sample volume, the measurements were performed using the rheometer instead [13]. A small amount of sample (~0.5–1.0 g) was loaded into the lower plate, and the gap between the two plates was closed at 2 mm by controlling the normal force. The gap was then closed from 2 mm to 0.5 mm at a constant speed of 0.1 mm/s by arresting the rotary motion of the upper plate. Finally, the upper plate was retracted to a gap of 3 mm at 0.1 mm/s.

Tribological measurements were performed using an in-house linear reciprocating tribometer at a load of 0.5 N and at a sliding speed of 10 mm/s. The friction measurements were performed on Bioskin (Beaulax, Co., Ltd., Saitama-shi, Saitama prefecture, Japan) to avoid variation in actual skin tissue. Bioskin is made of urethane elastomer that has been processed to reproduce human skin characteristics. Frictional resistance of different ointments was measured at 25 ± 0.5 °C and RH 55 ± 3% by sliding a 25 mm diameter hemispherical PEEK probe against a Bioskin plate. For each measurement, about 0.4 g of sample was used, and the total displacement of the probe was limited to 30 mm. At least three measurements were performed for each sample. The readers are referred to Cyriac et al. [13,14] for more information on the methodology used.

#### 2.3.5. Thermal Analysis

Digital scanning calorimetry thermograms were acquired using a Mettler Toledo DSC3 calorimeter (Mettler Toledo, Singapore). The instrument was calibrated using indium as a standard prior to measurements. An appropriate amount of sample (5 mg) was placed in a crimpled aluminum pan and heated at a rate of 1 °C/min from 25 to 80 °C after equilibrating at 25 °C for 10 min, then subsequently cooled to 25 °C at the same rate. Two cycles of measurements were performed for each sample (1st heating→ 1st cooling→ 2nd heating→ 2nd cooling). The glass transition temperature (T_g_), melting temperature (T_m_), and melt crystallization temperature (T_c_) of petrolatum ointment samples were determined using STAR^e^ SW V14.00 analysis software, version 14.00. The T_g_ was determined as the midpoint of the change in heat capacity of the sample, while T_m_ and T_c_ were determined as the onset temperatures. The samples were purged with a stream of nitrogen at 50 mL/min.

#### 2.3.6. High Performance Liquid Chromatography (HPLC) Assay of Lidocaine

The assay of lidocaine was conducted using an HPLC (1100 series, Agilent Technologies, Santa Clara, CA, USA) equipped with a ZORBAX Eclipse Plus C18 column (4.6 mm × 250 mm, 5 µm) (Agilent Technologies, Santa Clara, CA, USA) equilibrated at 25 °C. A mobile phase consisting of a mixture of acetonitrile:water (20:80 *v/v*) and 5 wt% acetic acid was used at a flow rate of 1 mL/min. The injection volume, retention time, and detection wavelength of lidocaine were 20 mL, 5.1–5.3 min, and 262 nm, respectively.

#### 2.3.7. In Vitro Release (IVRT)

In vitro release testing (IVRT) (membrane permeation) studies from the ointments (1 wt% of lidocaine dissolved in PG) were conducted on a Phoenix RDS automated diffusion testing system (Teledyne Hanson, Chatsworth, CA, USA). The system consists of a 6-cell dry heat block and an automated sampling system that allows in vitro release experiments to be performed on 6 samples simultaneously. Each 14 mL Franz cell was filled with a receptor medium consisting of 70:30 *v/v* ethanol:water solution to maintain sink condition. The receptor medium was maintained at 32 ± 1 °C by the dry heat block and continuously stirred at 200 rpm. Approximately 0.25 g of ointment was evenly spread on a dosage chamber, and a 0.45 µm nylon membrane (25 mm disc) was placed in between the dosage chamber and receptor medium for in vitro release experiments. A glass cover was placed on top of the dosage chamber to seal the cell. Sample aliquots of 0.2 mL were withdrawn from the cell receptor compartment through the sampling port at 1 h intervals and analyzed for lidocaine concentrations using the HPLC method described previously.

### 2.4. Statistical Analysis—PCA

Principal component analysis (PCA) was performed using MATLAB to understand the relationship between operating process parameters and the measured properties of the ointment samples. PCA is a multivariate method that reduces the dimensionality of the dataset to allow visualization of the correlations among different variables.

## 3. Results and Discussion

### 3.1. Preliminary Experiments

Due to the complexity of the manufacturing process with many process parameters, several sequential DoE studies had to be executed. The goals of the initial DoE studies were to narrow down the range of processing parameters to be investigated for significant effects on the products. This was followed by a more detailed DoE on specific stages after evaluating the preliminary experimental data.

Runs OT1, OT2, and OT3 (see Table 3 for the operating conditions) were conducted to investigate the effect of homogenization speed. The homogenization speed [F_2_] was varied from 2500 to 8000 rpm while the cooling rate [F_1_] was set at 0.8 °C/min, and the compounding shear temperature [F_3_] was fixed at 43 °C. The maximum homogenization speed at the final cooling stage [F_4_] was limited to 5000 rpm because the mixture became highly viscous at that stage, and higher homogenization would result in a significant temperature rise in the mixture. No significant temperature rise was detected in the mixture inside the main vessel, even at the highest homogenization speed and longest duration applied in this study.

The microstructures of the ointments were analyzed using a polarized light microscope. Two distinct structures can be observed in the microscope images shown in Figure 2. The bigger ones in the size range of tens of micrometers appear to be covered with crystalline material, while the smaller ones exhibit smooth surfaces and are less than 10 μm in diameter. The crystalline nature of the exterior of the bigger structures is confirmed by cross-polarized light in Figure 3. Monoglycerides are known to be surface active agents and have been reported to crystallize at oil/water and oil/air interfaces [3,6,15,16,17,18]. Therefore, we believe that the bigger structures are PG droplets being surrounded by MDG crystals, while the smaller ones are free PG droplets dispersed in the continuous phase of the ointment. For simplicity, we refer to the bigger crystalline structures as crystallites and the smaller ones as PG droplets in the rest of the paper. The size distribution of the crystallites and PG droplets for samples OT 1 to OT3 is shown in Table 3. Unfortunately, due to limitations in the microscopy technique here, it was difficult to accurately determine the crystallite and PG droplet size, leading to large variations in the measured data despite making 300 measurements for each sample. Hence, we have decided not to include crystallite and PG droplet size as critical quality attributes to be considered in the subsequent experiments.

The stability of the ointment was characterized by the LUMiSizer, based on the space-and time-resolved extinction profiles (STEP) technology, and its instability index was determined via SEPView 6 software. Figure 4 shows the evolution of light transmission with time for OT1 to OT3 measured by the LUMiSizer, together with the corresponding photographs of the samples after measurement. The transmission profiles showed that bleeding (syneresis) [19,20] is observed on the top part of all three ointments (OT1 to OT3). A larger amount of syneresis can be observed for OT3, while the extents of syneresis for OT1 and OT2 are comparable. This suggests that OT3 is less stable than OT1 and OT2, which is also reflected by their respective instability indices in Table 3. The instability index of OT3 is almost double that of OT1 and OT2, suggesting that homogenization speeds of 8000 rpm [F_2_] and 5000 rpm [F_4_] have an adverse effect on the ointment stability. The top parts of the samples were extracted and verified to be part of the oil phase rather than propylene glycol, by a simple miscibility test.

Oscillatory rheology measurements were performed to study the viscoelastic behavior of the ointments. Storage modulus (G′) and loss modulus (G″) were calculated from the small angle oscillatory shear (SAOS) measurements. The complex shear modulus (G*), which is an indication of the material stiffness as a function of strain, was then computed by the equation below:$$\left(G^{*}=\sqrt{{\left(G^{\prime }\right)}^{2}+{\left(G^{\prime \prime }\right)}^{2}}\right)$$

Appendix A shows the complex modulus and stress as a function of strain for one of the ointment samples. The application of strain initially resulted in a linear response between stress and strain, and this is referred to as the linear viscoelastic regime (VLE). With further increase in strain, G′ and G″ crossed over and beyond crossover stress (σ_cross_), the ointment lost its structural integrity, and the viscous characteristics dominated over the elastic response. The viscoelastic parameters obtained for OT1-3 are shown in Table 3. The complex shear modulus of OT3 is significantly lower compared with those of OT1 and OT2, which are almost identical. This indicates that high homogenization speed resulted in ointment that is less stiff. Crossover stress values again suggest that high homogenization speed is not beneficial for attaining a robust microstructure. The lower crossover stress of OT3 indicates that the ointment loses its structural integrity at lower stress. OT2 exhibits the highest crossover stress, but interestingly, the crossover stress of OT1 is almost the same as that of OT3. This may suggest that a homogenization speed above 2500 rpm is required.

From the results of OT1-OT3, the operating conditions at OT3 yielded the most drastic changes to the properties of the ointment compared to OT1 and OT2. Therefore, the subsequent set of experiments (OT4-OT9) was designed based on OT3 by fixing [F_4_] at 5000 rpm to investigate if varying the cooling rate [F_1_] and homogenization speed [F_2_] will shift the formulation properties to the desirable range. The compounding shear temperature [F_3_] was set at 40 °C, which is the lower limit set by the manufacturer. During PG addition, homogenization was applied twice for 20 min duration, with 30 min between each application. The operating conditions of OT4 to OT9 are shown in Table 4. At the same homogenization speed, the cooling rate did not seem to affect the rheological properties G* and σ_cross_ (OT4 vs. OT5, OT6 vs. OT7, and OT8 vs. OT9). Most importantly, OT4 to OT9 exhibited similar stability as assessed by the instability index measured by the LUMiSizer. From these findings, we believe that the homogenization speed of 5000 rpm or above at the final cooling step during the compounding stage may have damaged the microstructure built up in the previous steps, thus masking the effect of the other three operating parameters on the stability of the ointment. Hence, lower homogenization speeds should be employed at step 5 in subsequent experiments.

### 3.2. Final Design of Experiments

Based on the preliminary runs OT1-OT9, the homogenization speed at step 5 [F_4_] was capped at 2500 rpm in the final DoE. The cooling rate [F_1_] did not significantly affect the properties of the ointment, so the cooling rate [F_1_] was fixed at 0.8 °C/min, which is more practically achievable in the plant than the faster rate of 1.5 °C/min. Homogenization speed at steps 3 and 4 [F_2_] was varied from 2500 to 8000 rpm. During PG addition, the homogenization was applied twice for 20 min duration, with 30 min between each application. Propylene glycol addition temperature [F_3_] was assessed at two levels, 40 °C and 46 °C, which are the lower and upper limits of the manufacturing process. The operating conditions or factors of the final DoE are shown in Table 5.

#### 3.2.1. Stability of Ointment

The instability indices of the ointments produced in the final DoE are shown in Table 5. Comparing OS5-1 with OS5-4, OS5-2 with OS5-5, and OS5-3 with OS5-6, when propylene glycol was added at 40 °C [F_3_], the stability of the ointment increases (instability index decreases) when the homogenization speed at step 5 [F_4_] was lowered. An exception was when the homogenization speed at the previous steps [F_2_] was at 8000 rpm, [F_4_] no longer had any influence on the stability. This means that if the mixture is subjected to a high level of shear before or during the addition of propylene glycol, either the formation process of the microstructure is being hindered or the microstructures already formed are destroyed before reaching step 5; therefore, the damage cannot be reversed, even if no homogenization is applied at step 5. Even though our results suggest that no homogenization at step 5 is beneficial to the stability of the ointment, this is not practical in a manufacturing plant. Some degree of gentle homogenization is still required to ensure homogeneity within the mixing vessel and the final ointment product.

When the propylene glycol addition temperature [F_3_] was at 46 °C, the homogenization speed at step 5 [F_4_] did not affect the stability of the ointment since the instability indices were almost the same (OS5-7 vs. OS5-9, OS5-10 vs. OS5-12). Similar to the observation when [F_3_] was set at 40 °C, at the same homogenization speed at step 5, the instability index increased significantly when the homogenization speed [F_2_] was increased from 2500 to 8000 rpm. In addition, the mixing duration of the PG addition step also influenced the stability of the ointment. The stability of the ointment was improved when the mixing duration was reduced from 60 min to 30 min.

Comparing OS5-1 with OS5-7, OS5-3 with OS5-8, and OS5-4 with OS5-9, the ointments are more stable when PG addition took place at 40 °C than at 46 °C.

#### 3.2.2. Rheological Properties of Ointment

From the rheological properties listed in Table 5, in general, both complex shear modulus and crossover stress increase with a decrease in step 5 homogenization speed [F_4_]. This can be better visualized graphically in Figure 5. A lower homogenization speed at step 5 improves the resistance of the ointment from structural changes when subjected to shear. Going from OS5-1 to OS5-3 and OS5-4 to OS5-6, both G* and σ_cross_ decrease with increasing homogenization speed [F_2_], regardless of the homogenization speed at step 5. This shows that shear at the steps before step 5 also plays a part in controlling the rheological properties of the ointment. The same can also be observed for samples OS-7 to OS-12.

To understand why step 5 heavily influences the rheological properties and stability of the ointment, thixotropy tests were performed on petrolatum, which is the dominant component in the formulation. The tests were conducted using a rough parallel plate geometry with a 1 mm gap between plates. The shear rate was increased from 0 to 100 s^−1^ and then reduced back to 0 using a stepwise shear ramp. Figure 6 shows the shear stress response to the change in shear rate during the thixotropy tests. It is obvious that the hysteresis area increases with decreasing temperature, which indicates that the petrolatum undergoes more severe deformation as the temperature decreases. The change in hysteresis area with temperature can be better visualized in Figure 7. The apparent viscosity values at 100 s^−1^ are also shown in Figure 7. Both the hysteresis area and apparent viscosity increased drastically when the temperature was decreased from 43 °C to 32 °C, which corresponds to the temperature change in step 5. This explains why shear at step 5 is a critical factor that influences the properties of the ointment. It is worth noting that the hysteresis area and apparent viscosity also increase significantly when the temperature was decreased from 50 °C to 43 °C, although not as drastically as from 43 °C to 32 °C. This explains why homogenization speed at the steps before step 5 also plays a role in influencing the rheological properties of the ointment. The temperature response during thixotropy tests is not surprising since the congealing point of petrolatum used here is 55 °C.

Figure 8 illustrates the effect of PG addition temperature on the rheological properties of the ointments, comparing OS5-1 with OS5-7, OS5-3 with OS5-8, and OS5-4 with OS5-9. Both complex shear modulus and crossover stress decreased with increasing PG addition temperature, regardless of homogenization speed at step 5. The ointments were also more stable when PG was added at 40 °C than at 46 °C, regardless of the step 5 homogenization speed, as discussed in the previous section. This suggests that the ability to retain structural integrity when subjected to shear corresponds to improved stability of the ointment.

The effect of PG addition mixing duration at 46 °C on complex shear modulus and crossover stress is shown in Figure 9. A shorter mixing duration clearly resulted in more rigid (higher complex modulus) and stronger (higher crossover stress) ointments. This also suggests that 30 min of mixing is sufficient for PG droplets to be sufficiently dispersed in the bulk phase. Further mixing may destroy the paraffin network structure that is starting to build up, as discussed in the analysis of the thixotropic measurement of petrolatum, which, in turn, leads to lower complex shear modulus and crossover stress.

#### 3.2.3. Sensorial Properties of Ointment

Figure 10 shows a typical set of data obtained from textural analysis. From the plot of load vs. time, the different sensorial attributes, including firmness, adhesive force, and stringiness, were evaluated. Firmness is defined as the maximum positive force required for a probe to penetrate to a predefined distance in the sample. Adhesive force is the maximum negative force needed to withdraw the sample, and stringiness is defined as the degree to which a product hangs on the probe when the probe is retracted. Spreadability was quantified as the area under the force–time curve during the vertical downward movement of the upper plate of the rheometer. Cohesion strength, defined as the force with which the material resists vertical upward movement after the sample was compressed, was quantified as the maximum force measured during the retraction cycle. The pick-up characteristics of the ointment are reflected by firmness, adhesion strength, and stringiness work done, while the rubout characteristics are represented by spreadability and cohesion strength.

The sensorial attributes measured for ointments in the final DoE are shown in Table 6. Graphical comparisons of the various textural properties are shown in Appendix A. All the textural properties measured increase with decreasing homogenization speeds at step 5 [F_4_] (Appendix A). Going from OS5-1 to OS5-3 and OS5-4 to OS5-6, all the pick-up characteristics decrease with increasing homogenization speed [F_2_], regardless of the homogenization speed at step 5. This shows that an increase in homogenization speeds [F_4_] and [F_2_] resulted in ointments that are less firm, less adhesive, and easier to pull apart. The rubout characteristics follow the same trend, which indicates that the ointments are more resistant to spreading at lower homogenization speeds. This agrees with our rheological measurements that the ointments are more resistant to structural change when subjected to shear when produced at a lower homogenization speed. Similarly, the textural properties were found to decrease with an increase in PG addition temperature (Appendix A), which again agrees with the rheological measurements. The effect of PG addition mixing duration at 46 °C on textural properties is shown in Appendix A. A shorter mixing duration yielded ointments with higher firmness, adhesion strength, stringiness work done, spreadability, and cohesion strength. Clearly, the trend observed in rheological properties (G* and σ_cross_) is the same as that observed in textural properties of the ointments. This similarity is because textural properties are largely dependent on the bulk rheological properties of the ointments.

The coefficient of friction (CoF) as a function of probe displacement on the bioskin plate without the application of ointment is shown in Appendix A. The results show that dynamic friction is preceded by a static regime, where the latter is the friction that prevents an object from moving when it is standing still. Since the dynamic friction coefficient is more relevant to skin feel, only the coefficients of friction in this regime are averaged to obtain the average coefficients of friction listed in Table 6. Comparing the CoF measured without the application of ointment (Appendix A) and the CoF values of the ointments in Table 6, the ointments can effectively provide a lubricating effect on the skin by lowering the CoF between the skin and the probe by multi-fold. The effects of step 5 homogenization speed [F_4_], PG additional temperature [F_3_], and PG addition mixing duration on the CoF are illustrated graphically in Appendix A. The results show that the homogenization speed at step 5 influences the friction characteristics of the formulation. A decrease in frictional resistance can be seen for the formulations that were subjected to higher homogenization speed, except for OS5-1. This discrepancy may be attributed to several factors, such as change in hydration, displacement of ointment by the probe, change in adhesiveness over time, etc. However, this has not been explored further. It is known that the magnitude of drag experienced by the topical formulations is reflective of its rheological properties. However, our data show that the bulk rheological properties do not directly correlate with the frictional characteristics of the ointments formulated at different compounding temperatures or for different PG mixing durations. This is contrary to the correlation observed between the textural attributes and rheological properties. This discrepancy is plausible, as the frictional response is not only a function of bulk fluid properties but is also characteristic of the thin film formed at the contact interface. A lower friction observed for OS5-2, OS5-3, and OS5-4 suggests that these formulations may be perceived as ‘greasy’ by the consumers. On the contrary, OS5-5, OS5-6, OS5-7, etc., with higher friction values, will be perceived as less greasy.

#### 3.2.4. In Vitro Release

A nylon membrane was used in the current study, as it has been shown to be inert, and lidocaine absorption on the membrane is minimal [21]. A 70:30 *v/v* ethanol:water solution was chosen as the receptor medium to ensure that the solubility of lidocaine is high enough to provide sink condition for the release studies. In vitro release studies were performed using the most stable (OS5-4) and least stable (OS5-8) ointments in the final DoE to investigate if the stability of the ointment would affect the drug release performance. The assumption is that the less stable formulation is due to inadequate microstructure and not appropriate rheological properties. Lidocaine, a local anaesthetic agent, was chosen as the model drug for this study. It was predissolved in propylene glycol and incorporated into the ointment during PG addition at step 4 of the compounding stage (Figure 1). Figure 11 shows the release profile of lidocaine from OS5-4 and OS5-8. Despite the large difference in instability indices, rheological properties, and sensorial properties, the two ointments unexpectedly did not show a significant difference in in vitro release performance. The release profiles in Figure 11a show no difference in the release rates between the two ointments. However, the release data fit well to the Higuchi model (Figure 11b), suggesting that the ointments behave as a matrix, with drug diffusion through the matrix controlling the release rate. When an ointment is applied to the skin, the active would have to diffuse from the dissolved state in a dispersed solvent droplet through the bulk matrix to the skin surface before penetrating through the skin. The diffusion rate of the active is expected to be influenced by the microstructure of the ointment, e.g., the location of the active molecule and how the active molecules are encapsulated, and/or the viscosity of the ointment. The more viscous the ointment is, the higher the barrier to the diffusion of the active through the bulk matrix [any citation]. This is clearly not observed for the ointments in this work. OS5-4 is much stiffer compared to OS5-8, yet the release behavior is almost the same. There are two possible locations where lidocaine can be found in the ointment: inside PG droplets encapsulated by crystallized MDG (termed crystallites here) or free PG droplets. The crystallites are much larger in size than the free PG droplets, so the release of lidocaine from the crystallites will not be dominant unless the ointment is sufficiently sheared to destroy/rupture the crystallized MDG shell. Therefore, the lidocaine release during in vitro release experiments should predominantly come from the free PG droplets. Even though there is large variation in the PG droplet size measurements, it can still be seen that the size of the PG droplets is relatively consistent at around 4 µm for all the ointment samples in the final DoE (Table 5). Also, lidocaine has very limited solubility in the continuous petrolatum phase compared to PG [22]; hence, release may be by direct diffusion from PG droplets into the receptor phase. In this scenario, bulk viscosity and the size of MDG crystallites may not have an influence on in vitro release. This is possibly the reason for the similar release performance observed for both ointments with different viscosities. This indicates that even though processing parameters have influence on quality attributes such as rheology, they have no influence on performance attributes such as in vitro release. This finding shows that the manufacturing process for this ointment is robust in delivering drug product with consistent release performance, even if processing parameters are varied. Since the release performance is insensitive to variation in processing parameters within the wide range studied, the rheological and sensorial properties can be tuned to tailor to the patient’s preferences by adjusting the processing parameters within the bounds set in the final DoE.

### 3.3. PCA Analysis

Given the large number of measured properties and operating conditions, PCA analysis was performed to understand how the various operating parameters are correlated to the measured properties. The scree plot shown as the inset in Figure 12 shows five principal components are required to account for more than 95% of the variance. However, the first two components, which describe 75% of the variance, are sufficient for the purpose of this work for a descriptive analysis of the correlation among the different variables. From the PCA biplot, homogenization speed at step 5 [F_4_] is highly correlated with the instability index, as the vectors form the smallest angle. This agrees with our previous analysis that [F_4_] is the critical operating parameter that determines the stability of the ointment. The higher the homogenization speed at step 5, the higher the instability index, and the less stable the ointment. Therefore, a lower homogenization speed at step 5 should yield more stable ointment. Homogenization speed at steps 2 and 4 [F_2_] also correlates positively with the instability index, although the influence is lower than [F_4_] since the angle between the instability index and [F_2_] vectors is larger than that between the instability index and [F_4_]. Overall, a lower homogenization speed at the different steps during the compounding stage is beneficial to obtaining a more stable ointment product. Homogenization speed at step 5 [F_4_] also correlates positively with the coefficient of friction but to a lesser extent than with the instability index. However, the homogenization speed at steps 2 and 4 [F_2_] is almost orthogonal with the coefficient of friction, indicating that there is no correlation between the two. In contrast, both homogenization speeds ([F_2_] and [F_4_]) are negatively correlated with linear rheological response and the measured textural properties. This suggests that lower homogenization speeds should result in ointments that are more rigid and less susceptible to shear damage (higher complex shear modulus and crossover stress) while giving a skin feel of being more adhesive, firmer, and more cohesive and spreadable. It is clear from the biplot that the textural properties are correlated positively with the linear rheological response (G*, s_cross_), as their vectors form acute angles. Among the textural properties, adhesive force is most highly correlated with complex shear modulus and crossover stress.

PG addition temperature [F_3_] is also positively correlated with the instability index, although not as highly correlated as the homogenization speed, since the angle between the instability index and PG addition temperature is larger than the angles between the instability index and homogenization speeds. Unlike homogenization speed, PG addition temperature correlates highly with the coefficient of friction. Hence, if PG is added at a lower temperature, a more stable and lubricating ointment can be achieved, although an ointment with a lower coefficient of friction may also be perceived as greasy during application on skin. PG addition temperature does not have much influence on the rheological properties and the pick-up characteristics (adhesive force, stringiness, and firmness), although it correlates negatively, to a certain extent, with the rubout characteristics (cohesion strength and spreadability).

PG addition mixing time does not have much influence on the instability index since the vectors are almost orthogonal with each other. This is different from our previous analysis that the instability index decreased when the mixing time was reduced from 60 min to 30 min. This discrepancy could be due to the limited amount of data (six sets, OS5-7 to OS5-12) that were also performed at a less favorable temperature of 46 °C, where the adverse effect of high temperature of solvent addition swamped any adverse effects of additional mixing time. Nevertheless, the influence of PG addition mixing time is still less significant compared to homogenization speeds.

### 3.4. Stabilization Mechanism

From the final DoE, we found that ointments were more stable if PG was added at 40 °C than at 46 °C, regardless of the homogenization speeds applied at different stages. To explain this observation, DSC experiments were performed on the constituent components and combinations of the components. The thermograms obtained are shown in Figure 13. Only glass transition was observed for petrolatum and a mixture of petrolatum and wax. However, a broad melting peak during heating and two crystallization peaks (onset temperatures: 56.4 °C and 57.9 °C) during cooling were observed for pure MDG. The two crystallization peaks can be attributed to monoglycerides and diglycerides in MDG, with the monoglycerides crystallizing at a higher temperature than the diglycerides [3]. When MDG was added to petrolatum, two melting peaks and two crystallization peaks were observed. The crystallization onset temperatures were lowered to 51.8 °C and 44.2 °C. Since only glass transition occurred in petrolatum, the crystallization peaks observed can be attributed to MDG crystallization. DSC was also performed on the ointment samples, and the thermograms for two of the ointments are shown in Figure 14. In the presence of more components, the crystallization temperature of MDG was further lowered to around 36 °C. During DSC experiments, the samples were placed in a crucible and unstirred throughout. However, the mixtures are stirred and homogenized during the ointment preparation process; the crystallization of MDG is likely to occur earlier during cooling, i.e., at a temperature higher than observed in DSC since agitation will lower the activation energy required for crystal nucleation and facilitate faster and earlier nucleation [23,24,25,26,27].

Based on the DSC results, we propose a hypothesis as illustrated in the schematic shown in Figure 15. When the continuous phase is cooled from 75 °C to 40 °C, MDG may start to crystallize. When PG is added and the mixture is stirred and homogenized, the monoglyceride (MG), the components of MDG, and crystals move to the vicinity of the PG droplets, and encapsulation of PG droplets by MG crystals starts to take place while diglyceride (DG) forms a crystal network in the bulk. Upon further cooling to the final temperature of 32 °C, the PG droplets are effectively encapsulated by MG crystals, which provides stability to the ointment. On the other hand, if the mixture is cooled to 46 °C, MG crystallization may not have started, or only very few MDG crystals are nucleated. As a result, there are no or insufficient MDG crystals present during PG addition to begin the encapsulation process. Upon further cooling and dispersing, MDG crystals start to nucleate, but the mixture becomes too viscous for the MDG crystals to diffuse to the surface of the PG droplets for effective encapsulation. Therefore, the resulting ointments are less stable compared to when PG was added at 40 °C. Ali et al. [3] have shown that monoglyceride (MG) crystals have higher affinity to liquid–liquid interface in emulsions than diglyceride (DG) crystals. Monoglyceride crystals form shells around polyethylene glycol (PEG) droplets, while diglyceride crystals form a network-like structure in the bulk domain. They have shown similarly that a higher temperature for addition of the solvent tends to produce a less stable emulsion due to the inability to provide stabilization by the Pickering mechanism for emulsion.

The effect of step 5 homogenization speed [F_4_] can also be explained by our proposed hypothesis. In step 5, the MDG crystals begin to crystallize and encapsulate the PG droplets. Homogenization, subsequently, may disrupt the formation process of the microstructure or destroy the microstructure already formed. Therefore, we observed that a lower step 5 homogenization speed resulted in more stable ointment.

## 4. Conclusions

This study demonstrated the significant effects that selected processing parameters have on the critical quality attribute of ointment drug product. However, not all processing parameters are equal and have an equivalent significant effect on all or some critical quality attributes, including performance, stability, and sensorial properties. It appears that the temperature of addition of solvent and homogenization speed during cooling at the compounding stage had the most significant effect on rheological properties, stability, and sensorial properties. These were attributed to the complex dynamics of stabilizing agents (MDG) crystallizing at different temperatures during/after addition of solvent to enable droplet encapsulation and stabilization. Once the optimal microstructure is formed, additional homogenization (speed and time) may disrupt the structure to cause instability. Rheological properties and instability, as measured by the LUMiSizer, appear to correlate well and provide insight into the formed microstructure of the ointment DP. However, based upon similar in vitro release profiles for the suboptimal DP compared to the most stable DP, it appears that the variations in microstructure in the ranges affected by changes in the processing parameters studied may have no influence on delivery performance. This suggests that it is difficult to obtain an ideal product profile from key quality attributes, including performance, stability, and sensorial properties, by modifications of process alone. From a manufacturing scale-up perspective, we have demonstrated how a typical semi-solid manufacturing process can be studied and optimized to achieve the target product profile from a quality and performance attributes perspective.

## Figures and Tables

**Figure 1 pharmaceutics-15-02219-f001:**
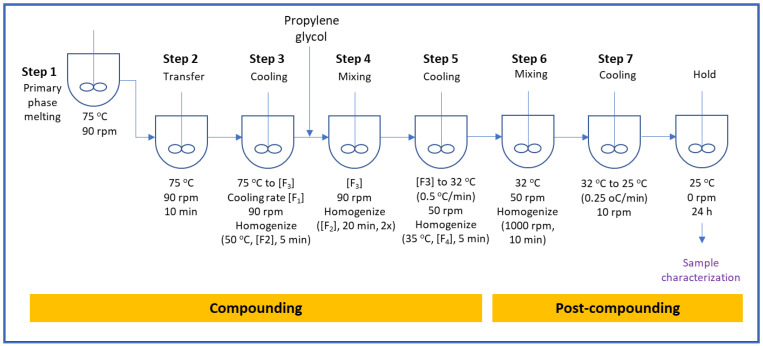
Ointment process flow diagram with the set points for temperature and process parameters for the DoE studies.

**Figure 2 pharmaceutics-15-02219-f002:**
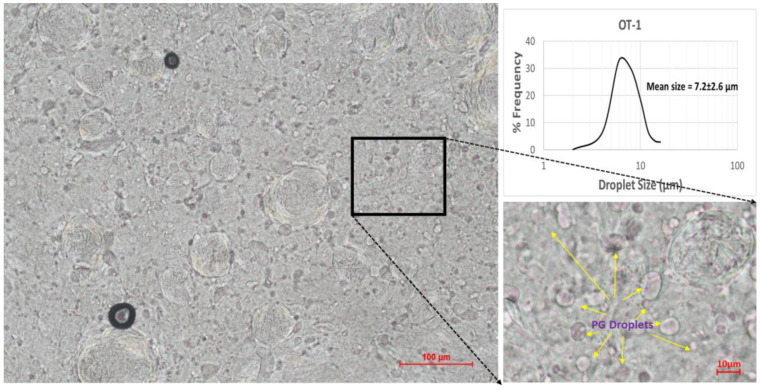
Microscope images illustrating how PG droplets are identified. Image on the right is magnified from the boxed region of the left image. Yellow arrows show some of the PG droplets.

**Figure 3 pharmaceutics-15-02219-f003:**
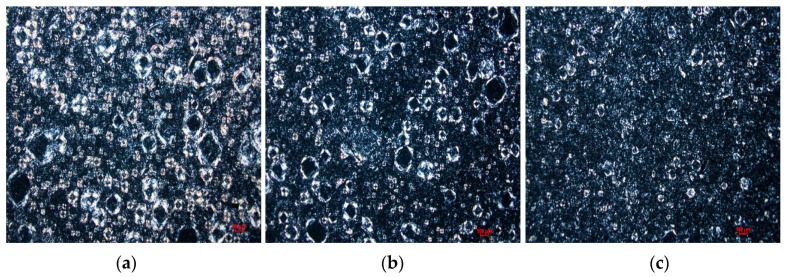
Polarized microscope images showing crystallites observed in samples (**a**) OT-1, (**b**) OT-2, and (**c**) OT-3.

**Figure 4 pharmaceutics-15-02219-f004:**
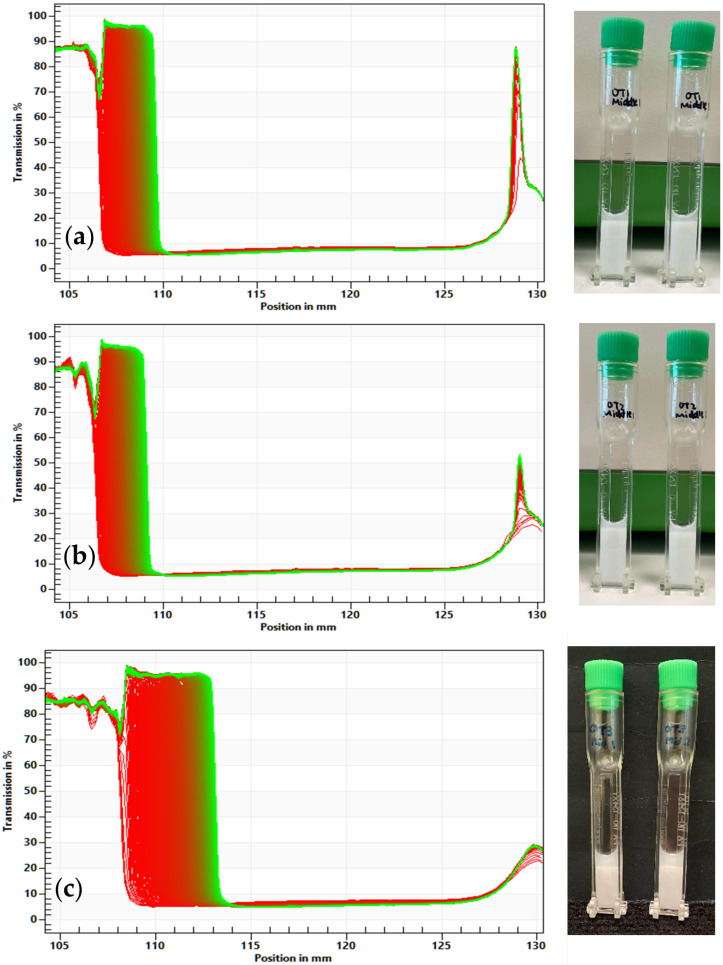
Evolution of light transmission with time over the length of the sample (**a**) OT1, (**b**) OT2, and (**c**) OT3, measured by the LUMiSizer, and the corresponding photographs of the samples after measurement. The *y*-axis of the graph is % light transmission, and the *x*-axis is the vertical position in mm. In each graph, red lines are the transmission data of the first and subsequent scans while green line is the transmission data of the last scan.

**Figure 5 pharmaceutics-15-02219-f005:**
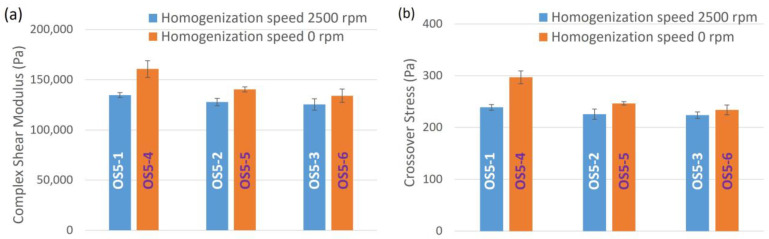
Effects of homogenization speed at STEP 5 [F_4_] on (**a**) complex shear modulus and (**b**) crossover stress.

**Figure 6 pharmaceutics-15-02219-f006:**
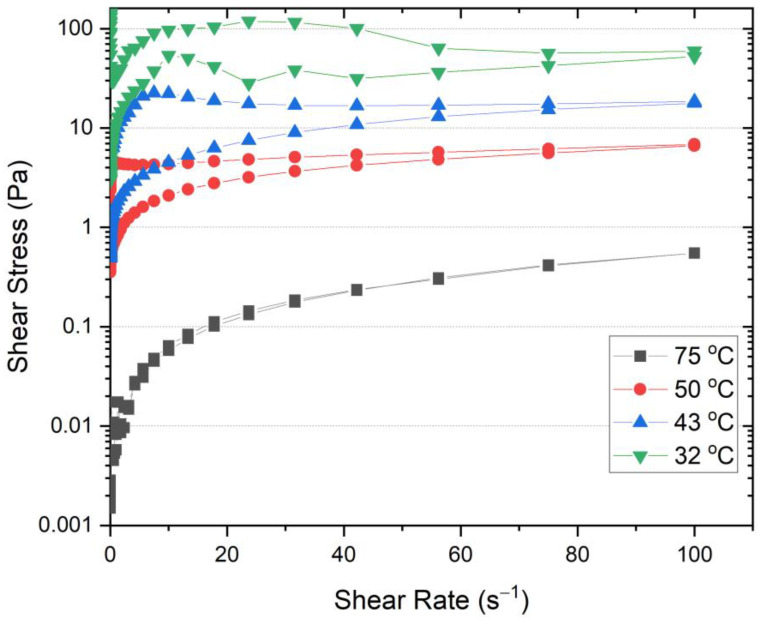
Thixotropic measurement of petrolatum at different temperatures.

**Figure 7 pharmaceutics-15-02219-f007:**
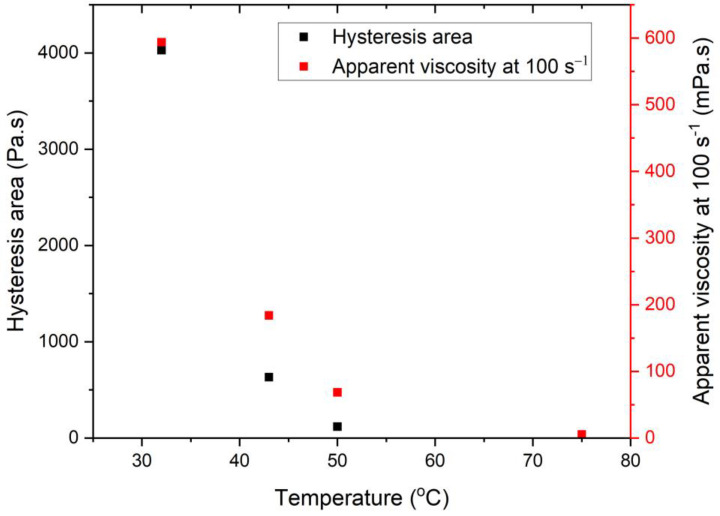
Changes in hysteresis area and apparent viscosity at 100 s^−1^ with temperature for thixotropy measurements of petrolatum.

**Figure 8 pharmaceutics-15-02219-f008:**
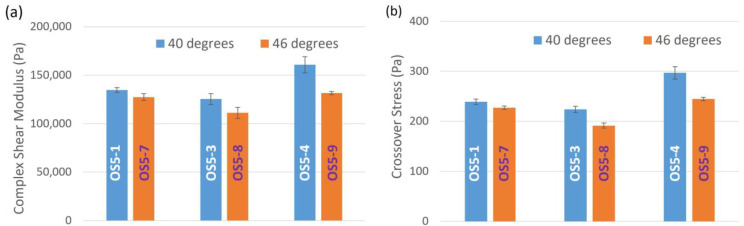
Effect of PG addition temperature [F_3_] on (**a**) complex shear modulus and (**b**) crossover stress.

**Figure 9 pharmaceutics-15-02219-f009:**
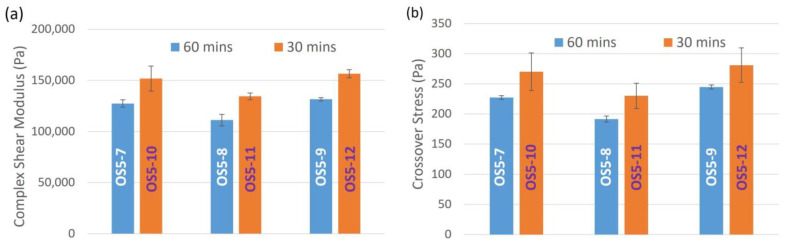
Effect of PG addition mixing time at 46 °C on (**a**) complex modulus and (**b**) crossover stress.

**Figure 10 pharmaceutics-15-02219-f010:**
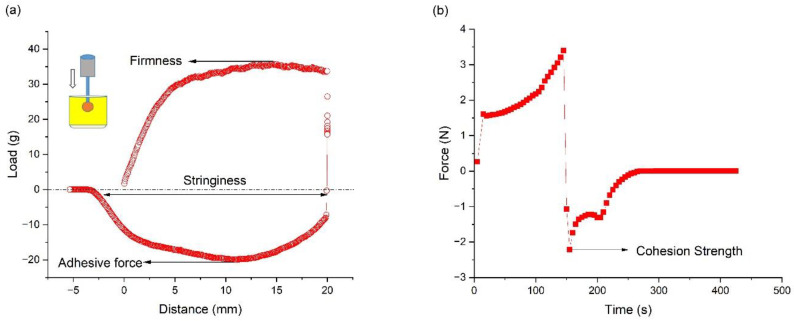
(**a**) Load–distance curve obtained from texture analysis measurement and (**b**) spreadability and cohesion strength measurement data obtained using a rheometer for OS5-8.

**Figure 11 pharmaceutics-15-02219-f011:**
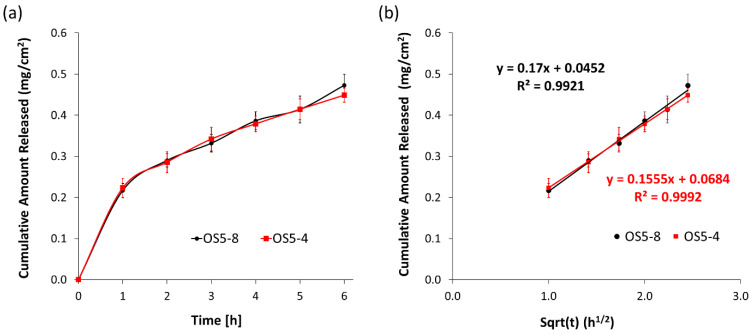
(**a**) Cumulative release of lidocaine from ointments OS5-4 and OS5-8; (**b**) cumulative release data fitted to the Higuchi model.

**Figure 12 pharmaceutics-15-02219-f012:**
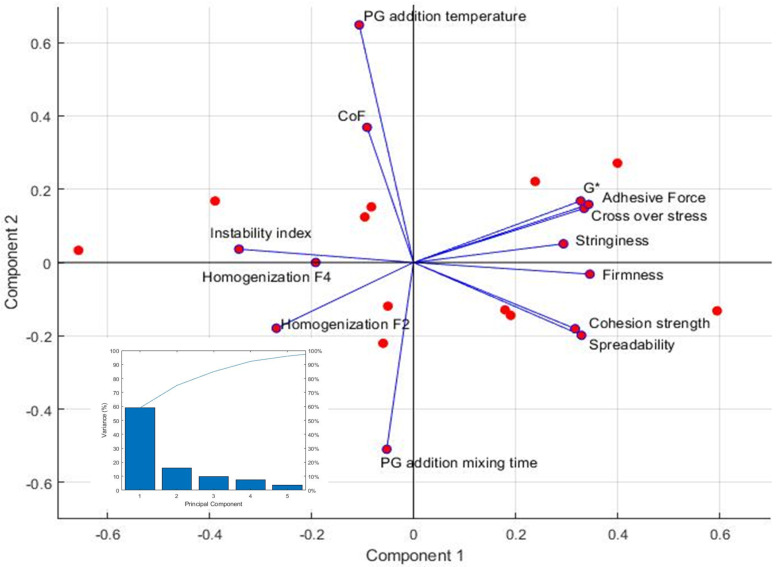
PCA biplot obtained by projecting the variables onto the first two principal components. Inset shows the scree plot.

**Figure 13 pharmaceutics-15-02219-f013:**
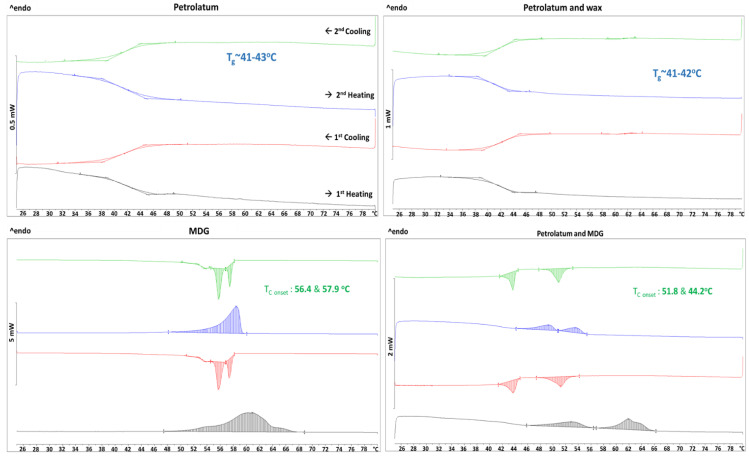
DSC thermograms of petrolatum, petrolatum + wax, MDG, and petrolatum + MDG.

**Figure 14 pharmaceutics-15-02219-f014:**
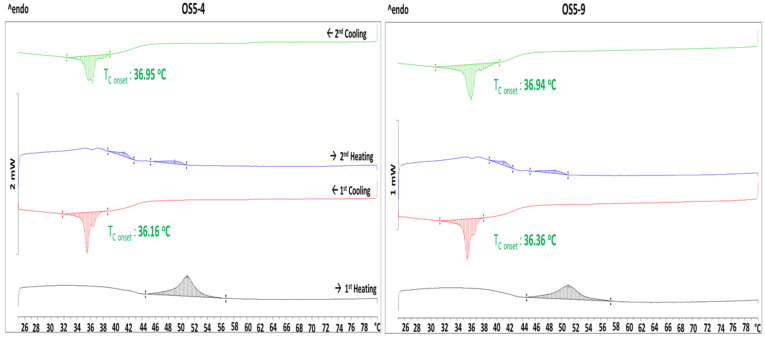
DSC thermograms of OS5-4 and OS5-9.

**Figure 15 pharmaceutics-15-02219-f015:**
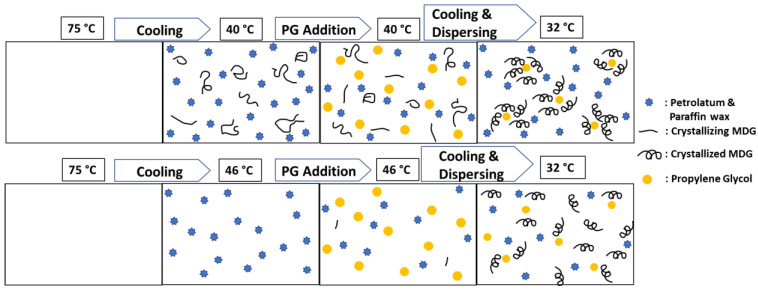
Schematic of MDG crystallization and distribution at different cooling temperatures.

**Table 1 pharmaceutics-15-02219-t001:** Ointment composition.

Ingredient	Phase	Wt.%
Propylene glycol (PG)	Solvent for active and hence referred to as active phase	9
White petrolatum (WP)	Primary phase or continuous phase	79
Mono-and-Diglycerides (MDG)		7
Paraffin wax (PW)		5
**Total**		**100**

**Table 2 pharmaceutics-15-02219-t002:** Levels of each processing condition studied.

	Processing Conditions	Levels
[F_1_]	Cooling rate	0.8 ± 0.1, 1.5 ± 0.1 °C/min
[F_2_]	Homogenization speed (compounding stage)	2500, 4000, 8000 rpm
[F_3_]	Active phase (solvent) addition temperature	46 ± 2, 43 ± 2, 40 ± 2 °C
[F_4_]	Compounding final cooling stage homogenization speed	2500, 5000 or 0 rpm

**Table 3 pharmaceutics-15-02219-t003:** Dry run: Operating conditions, LUMiSizer instability index, PG droplet, crystallite size, and viscoelastic properties for dry runs OT1 to OT3.

Compounding Stages	Conditions	Sample		
		OT1	OT2	OT3
Cool 75 °C to 43 °C (Primary phase melting)	Homogenization speed [F_2_] (rpm)	2500	4000	8000
	Cooling rate [F_1_] (°C/min)	0.8	0.8	0.8
Addition of PG at 43 °C (Active phase distribution)	Homogenization speed [F_2_] (rpm)	2500	4000	8000
	Homogenization frequency	4 times for 5 min	2 times for 10 min	2 times for 20 min
	Homogenization interval	Every 15 min	Every 30 min	Every 30 min
	Total mixing and homogenization time	60 min	60 min	60 min
Cool 43 to 32 °C (Final Cooling Compounding Stage, STEP 5)	Homogenization speed [F_4_] (rpm)	2500	4000	5000
Microscopy measurements	Crystallite size (µm)	84.5 ± 56.5	71.7 ± 51.1	55.4 ± 34.0
	PG droplet (µm)	7.2 ± 2.6	6.7 ± 2.6	6.9 ± 2.6
Instability Index (IS)		0.149 ± 0.009	0.135 ± 0.004	0.246 ± 0.001
Complex Shear Modulus (G*) (Pa)		144,193 ± 3016	144,064 ± 3749	104,136 ± 372
Crossover Stress (σ_cross_)(Pa)		260 ± 5.7	334 ± 1.4	261 ± 0.0

**Table 4 pharmaceutics-15-02219-t004:** Preliminary DoE: Operating conditions, LUMiSizer instability index, PG droplet, crystallite size, and viscoelastic properties for OT4 to OT9.

Compounding Stages	Conditions	Sample					
		OT4	OT5	OT6	OT7	OT8	OT9
Cool 75 to 40 °C (Primary Phase)	Homogenization speed [F_2_] (rpm)	2500		4000		8000	
	Cooling Rate [F_1_] (°C/min)	0.8	1.5	0.8	1.5	0.8	1.5
Addition of PG at 40 °C (Active Phase)	Homogenization speed [F_2_] (rpm)	2500		4000		8000	
	Cooling Rate [F_1_] (°C/min)	0.8	1.5	0.8	1.5	0.8	1.5
Cool 40 to 32 °C (Final Cooling Compounding Stage, STEP 5)	Homogenization speed [F_4_] (rpm)	5000					
	Cooling rate (°C/min)	0.5					
Microscopy measurements	Crystallite size (µm)	60.8 ± 39.4	61.7 ± 42.2	49.5 ± 24.8	52.1 ± 22.7	49.0 ± 16.2	50.0 ± 19.0
	PG Droplet (µm)	4.6 ± 2.3	4.4 ± 2.3	4.4 ± 2.2	4.1 ± 1.8	4.2 ± 1.8	4.8 ± 1.7
Instability Index (IS)		0.316 ± 0.001	0.306 ± 0.001	0.312 ± 0.007	0.302 ± 0.006	0.325 ± 0.006	0.303 ± 0.001
Complex Shear Modulus (G*) (Pa)		134,689 ± 1614	129,826 ± 4622	131,930 ± 8524	143,656 ± 208	103,486 ± 5604	106,269 ± 8579
Crossover Stress (σ_cross_) (Pa)		282 ± 3	293 ± 11	274 ± 7	271 ± 25	213 ± 29	212 ± 29

**Table 5 pharmaceutics-15-02219-t005:** Operating conditions, instability indices, and rheological properties for ointment samples in final DoE.

Compounding Stages	Run Conditions	Run/Sample											
		Run 1/OS5-1	Run 2/OS5-2	Run 3/OS5-3	Run 4/OS5-4	Run 5/OS5-5	Run 6/OS5-6	Run 7/OS5-7	Run 8/OS5-8	Run 9/OS5-9	Run 10/OS5-10	Run 11/OS5-11	Run 12/OS5-12
Cool 75 °C to 40 or 46 °C (Primary Phase)	Homogenization speed [F_2_] (rpm)	2500	4000	8000	2500	4000	8000	2500	8000	2500	2500	8000	2500
Addition of PG at 40 or 46 °C (Active Phase)	Homogenization speed [F_2_] (rpm)	2500	4000	8000	2500	4000	8000	2500	8000	2500	2500	8000	2500
Cool at 40 or 46 °C to 32 °C (Final Cooling Compounding Stage, STEP 5)	Homogenization speed [F_4_] (rpm)	2500	2500	2500	0	0	0	2500	2500	0	2500	2500	0
Addition of PG	Temperature [F_3_] (°C)	40						46			46		
	Mixing Duration (minutes)	60						60			30		
Microscopy measurements	Crystallite size (µm)	63 ± 39	57 ± 32	51 ± 26	69 ± 36	67 ± 39	55 ± 30	63 ± 41	57 ± 28	58 ± 44	59 ± 48	56 ± 31	57 ± 46
	PG droplet (µm)	4.0 ± 2.1	4.0 ± 2.3	3.6 ± 2.1	3.8 ± 2.0	3.7 ± 1.9	4.1 ± 1.9	3.8 ± 1.9	4.3 ± 2.1	3.9 ± 2.1	3.6 ± 1.5	3.5 ± 1.5	3.5 ± 1.5
Instability Index (IS)		0.274 ± 0.006	0.276 ± 0.007	0.291 ± 0.002	0.260 ± 0.005	0.266 ± 0.009	0.293 ± 0.008	0.281 ± 0.001	0.314 ± 0.003	0.282 ± 0.001	0.270 ± 0.003	0.303 ± 0.007	0.266 ± 0.006
Complex Shear Modulus (G*) (Pa)		134,691 ± 2402	127,778 ± 3648	125,326 ± 5720	160,732 ± 8261	140,361 ± 2494	134,095 ± 6685	127,444 ± 3576	111,153 ± 5699	131,524 ± 1647	151,804 ± 12,294	134,379 ± 3348	156,473 ± 3891
Crossover Stress (σ_cross_) (Pa)		239 ± 6	226 ± 10	224 ± 6	297 ± 13	247 ± 3	234 ± 10	227 ± 3	191 ± 5	245 ± 4	243 ± 31	206 ± 21	248 ± 29

**Table 6 pharmaceutics-15-02219-t006:** Texture analyzer measurements and friction coefficients for OS5-1 to OS5-12.

Sample	Instability Index	Firmness (g)	Adhesive Force (g)	Spreadability (N·s)	Stringiness Work Done (mJ)	Cohesion Strength (N)	Friction Coefficient (CoF)
OS5-1	0.274	54.8 ± 3.2	28.8 ± 2.2	267 ± 2	2.1 ± 0.3	1.90 ± 0.02	0.053 ± 0.009
OS5-2	0.276	47.7 ± 3.6	24.9 ± 1.7	248 ± 11	1.6 ± 0.2	1.79 ± 0.09	0.042 ± 0.016
OS5-3	0.291	40.7 ± 1.8	21.9 ± 0.8	231 ± 7	1.6 ± 0.1	1.66 ± 0.07	0.060 ± 0.022
OS5-4	0.260	62.1 ± 0.9	31.2 ± 1.5	309 ± 9	2.2 ± 0.2	2.27 ± 0.06	0.046 ± 0.008
OS5-5	0.266	51.0 ± 1.8	26.4 ± 1.7	294 ± 5	1.7 ± 0.1	2.22 ± 0.01	0.079 ± 0.011
OS5-6	0.293	48.2 ± 2.0	25.4 ± 1.4	253 ± 0	1.8 ± 0.2	1.95 ± 0.13	0.095 ± 0.007
OS5-7	0.281	50.2 ± 3.6	27.2 ± 2.3	213 ± 1	2.0 ± 0.4	1.41 ± 0.06	0.103 ± 0.016
OS5-8	0.314	36.1 ± 0.8	20.1 ± 0.1	177 ± 8	1.4 ± 0.1	1.01 ± 0.10	0.091 ± 0.011
OS5-9	0.282	43.8 ± 2.7	25 ± 1.3	210 ± 4	1.8 ± 0.1	1.34 ± 0.04	0.086 ± 0.003
OS5-10	0.270	52.1 ± 1.1	29.4 ± 1.1	282 ± 16	1.7 ± 0.2	2.25 ± 0.09	0.079 ± 0.014
OS5-11	0.303	41.6 ± 1.3	22.2 ± 2.0	181 ± 2	1.5 ± 0.2	1.09 ± 0.02	0.062 ± 0.004
OS5-12	0.266	53.9 ± 4.3	31.8 ± 0.6	269 ± 20	2.1 ± 0.3	2.11 ± 0.17	0.081 ± 0.011

## Data Availability

The data that support the findings of this study are available from the corresponding author, Ron Tau Yee Lim (lim_tau_yee@isce2.a-star.edu.sg), upon reasonable request.

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
