# Peer review of "Influence of Manufacturing Process on the Microstructure, Stability, and Sensorial Properties of a Topical Ointment Formulation"

_pharmaceutics, 2023, doi:10.3390/pharmaceutics15092219_

Round 1
Reviewer 1 Report
Comments:
The effect of the manufacturing process on the microstructure, stability, and sensorial properties of a topical ointment formulation was evaluated in detail. It is a well designed study, and the results are discussed in detail. However, the following issues should be addressed by the authors to improve the quality of the manuscript.
- Abstract: results of the IVRT studies should also be stated in the abstract.
- Sample amounts used in characterization studies should be given in parentheses where appropriate (e.g. for physical stability, rheological properties, thermal analysis)
- It was stated that microstructures of the ointments were analyzed using a polarized light microscope. However, Raman or a high resolution microscopic analysis of microstructures of the ointments is more appropriate.
- IVRT studies:
o Selection of membrane and release media should be justified.
o Information on membrane integrity, membrane inertness, pH and duration of the release studies should be given.
o According to the regulatory guidelines (i.e. EMA), IVRT studies should be performed up to 70% of the drug released. Please give information about the released amount (%) at the end of the experiment. Author should also provide the release profiles as the cumulative amount released (%) vs squared root t.
- Statistical comparisons between the parameters were missing (e.g. Table 5). Any significant difference between the parameters should be given with the p value.
Reviewer 2 Report
This manuscript investigated the influence of processing parameters on the stability and sensorial properties of ointments by applying a Design of Experiments approach. The study identified critical processing parameters and determined the possible mechanisms for the observed effects. The research could demonstrate a Quality-by-Design method in processing design of ointments. Minor comments:
1. As a DoE approach was used, it would be suggested to add more details about the method, for example, design model, run pattern, run number, and run conditions for each run. Table 4 may be redesigned to add this information.
2. The levels of F1 to F4 (lines 136 to 141) may be presented in a table.
3. Section 2.2: please add how the drug was added.
4. Line 126: 36 deg C or 32 deg C?
5. The effects of processing parameters were investigated in this manuscript. The authors stated that “From a manufacturing scale-up perspective, …. (lines 665-668). Would it be possible to propose optimal processing parameters?
Reviewer 3 Report
In the present study, the authors have undertaken and investigation into the Influence of Manufacturing Process on the Microstructure, Stability, and Sensorial Properties of a Topical Ointment Formulation. This is achieved by means of a large DoE whereby the physico -chemical properties of candidate formulations microscopy, physical stability, rheology, texture & tribology, thermal analysis, chromatographic analysis and in vitro release was evaluated and statistically analysed by means of PCA. The manuscript is well-organized and carefully edited. However, several issues require attention and resolution before acceptance. These are highlighted below:
In general the temperature symbol used in the main body of the text is not correct. Consider using the windows shortcut (Alt+0176) to avoid the error in future.
Figure 2 - A new scale should be provided on the zoomed image
Line 303 - Thought should be given the placing the equation a standalone line.
The authors should provide a rationale for the DoE approach taken based on the typical attributes of a DoE study.
Figure 5 - Could statistical analysis (Anova) be used in this instance to demonstrate if there is a statistical difference observed from the treatments. This is not a requirement for publication merely a suggestion. Similar for Figures 8 and 9
